# Data Efficient Continual Learning of Large Language Model

## Abstract

Continual Learning (CL) in large language models (LLMs) aims to enable models to learn from evolving data distributions while preserving previously acquired knowledge. However, existing CL methods primarily rely on statistical correlations from observed data, which are particularly vulnerable under limited data settings. This reliance results in two major drawbacks: (1) increased susceptibility to forgetting previously learned knowledge when data distribution shifts occur, and (2) a tendency to depend on spurious features instead of uncovering true causal relationships in new tasks. These issues become even more pronounced, especially when training data is limited. To address these challenges, we introduce a causality-guided CL approach that reinterprets CL through the lens of causal inference. Our method aims to mitigate the dependency of model parameters on the data inputs, leading to two key advantages: (1) reduced catastrophic forgetting, and (2) decreased dependence on spurious correlations, thereby improving generalization across both old and new tasks. Extensive experiments on pre-trained LLMs, including T5-large and Llama2, demonstrate that our approach significantly outperforms state-of-the-art (SOTA) CL methods in LLMs, particularly when the amount of training data is limited.

## 1 Introduction

Large Language Models (LLMs) have demonstrated remarkable performance across a wide range of natural language processing (NLP) tasks, including text generation, translation, sentiment analysis, and question-answering (Brown et al., 2020; Achiam et al., 2024). These models have revolutionized the field by achieving human-like proficiency in many applications, driven by their ability to process and generate vast amounts of text with a deep understanding of context and semantics. However, despite these impressive achievements, a critical challenge remains: enabling LLMs to continually learn and adapt to new information while preserving the knowledge acquired from previous tasks. This capability is essential for the development of truly autonomous and intelligent systems, as it would allow LLMs to evolve over time Ke et al. (2023), expanding their knowledge base without suffering from catastrophic forgetting (CF) (McCloskey & Cohen, 1989) —a common problem where newly learned information interferes with previously acquired knowledge. Achieving this level of continual learning (CL) is a crucial step toward realizing artificial general intelligence (AGI), where a system can seamlessly integrate and apply knowledge across diverse domains, exhibiting a level of adaptability and intelligence comparable to human cognition.

Current CL methods for LLMs primarily focus on capturing statistical correlations between input data and labels. Although these approaches can be effective, it exposes critical vulnerabilities in retaining past knowledge and acquiring new knowledge. (1) Firstly, these methods are particularly susceptible to non-stationary data distributions during training. Most CL techniques are designed to reinforce statistical relationships observed in the training data, which often lack resilience when encountering non-stationary data distribution. This problem is especially pronounced in settings with limited labeled data. As training data distributions change, models that rely exclusively on statistical correlations struggle to generalize, resulting in degraded performance and intensified catastrophic forgetting. (2) Second, during the learning of new tasks, CL models often memorize spurious features instead of identifying true predictive patterns in the data (Bombari & Mondelli, 2024). This problem becomes more pronounced when training data is scarce, as the model may overemphasize coincidental patterns that appear to correlate with the output labels. In text classification, spurious

features are input patterns that seem related to the target label but lack a true causal connection to the task. These misleading features can cause the model to make errors, especially when the true task-relevant features differ. For instance, consider a sentiment analysis model classifying movie reviews as positive or negative. A spurious feature could be the word "Oscar," often seen in positive reviews but not necessarily a sign of sentiment. For example: "This movie was boring despite the Oscar nomination." (Negative) "An amazing Oscar-winning performance!" (Positive). The model might mistakenly learn that "Oscar" always implies a positive sentiment, leading it to misclassify the first sentence as positive despite its negative tone. Spurious features like this can make the model overfit to irrelevant correlations in the training data, causing a significant drop in performance when faced with new data that doesn't follow the same patterns. This challenge is worsened when training data is limited, making the model overly dependent on superficial associations rather than genuine predictive rules. Therefore, traditional CL approaches that rely on statistical correlations often struggle to retain past knowledge and adapt effectively to new tasks, leading to suboptimal performance.

Humans can learn efficiently from a limited number of examples by leveraging causal knowledge. Even with small amounts of data, understanding the causal relationships enables us to make accurate predictions and decisions. Causal inference aims to replicate this efficiency by identifying stable causal factors, which allows models to perform well even with limited data. Furthermore, humans intuitively understand that not all correlations imply causation and often disregard spurious relationships. Causal inference (Pearl, 2009), similarly, helps models avoid misleading correlations by adjusting for variables that might obscure the true causal effect.

Inspired by how humans process new information while retaining old knowledge, and recognizing the limitations of relying solely on statistical correlations in existing CL methods, we propose a novel approach that models and leverages the causal relationships among different random variables in CL for LLMs. Causal relationships go deeper than mere correlations. By identifying and modeling these causal connections, models can potentially become more robust to distributional shifts. They would not merely react to observed patterns but would instead infer the underlying mechanisms driving those patterns. By grounding CL models in causal reasoning, we can enhance their ability to generalize across different contexts. When the data distribution shifts, a model that understands the causal pathways can adjust more effectively rather than just memorizing surface-level correlations. This shift from a correlation-based approach to a causality-guided framework represents a fundamental change in how we approach CL in LLMs.

To achieve causal learning during CL in LLMs, we begin by analyzing the CL process through a causal lens, modeling the relationships among key variables. Our findings reveal that the heavy dependence of CL model parameters on input data is a critical factor contributing to two major issues: (1) catastrophic forgetting of previously learned knowledge, as over-reliance on data inputs causes significant parameter shifts when the training data distribution changes; and (2) the tendency to memorize spurious correlations in new tasks, leading to poor generalization. To address these challenges, we propose a soft intervention approach that mitigates the influence of input data on model parameters. By reducing the impact of data inputs, our method offers two primary benefits: (1) diminished catastrophic forgetting of prior knowledge, as the model parameters become less sensitive to shifts in data distribution; and (2) reduced memorization of spurious correlations when learning new tasks, enhancing the model's ability to identify true causal relationships. As a result, our approach significantly improves performance across both previously learned and newly introduced tasks, delivering a more robust and generalizable CL framework.

To assess the effectiveness of our proposed method against SOTA approaches, we conduct extensive experiments on multiple datasets and pre-trained LLMs, including T5-large and Llama2, under limited labeled data settings. The results demonstrate that our method significantly outperforms existing SOTA CL methods for LLMs.

Our contributions can be summarized as follows:

- We present a novel and general causal machine learning framework applicable to a wide range of machine learning models and tasks. In particular, we tailor this general framework to CL in LLMs. By reinterpreting the CL process in LLMs through the lens of causality, we offer a fresh perspective on mitigating catastrophic forgetting.
- We develop a causality-guided CL algorithm that reduces the dependence of model parameters on input data, achieving two key advantages: minimizing forgetting of previously

learned knowledge and enhancing new task performance by mitigating the memorization of spurious correlations.

- Through extensive experiments on multiple datasets and large-scale language models, we demonstrate that our method significantly outperforms SOTA CL approaches in LLMs.

## 2    RELATED WORKS

### 2.1    TRADITIONAL CONTINUAL LEARNING

Traditional CL approaches can be broadly classified into the following categories: (1) regularization-based methods (Kirkpatrick et al., 2017; Li & Hoiem, 2017; Zenke et al., 2017; Chaudhry et al., 2018), which constrain model updates to prevent rapid changes and preserve previously learned knowledge; (2) memory-replay-based methods (Chaudhry et al., 2019; Buzzega et al., 2020; Pham et al., 2021; Caccia et al., 2022; Arani et al., 2022; Yang et al., 2023; Wang et al., 2024b), which store a small subset of examples from prior tasks in a memory buffer and replay them during training; (3) architecture-based methods (Rusu et al., 2016; Li et al., 2019; Konishi et al., 2023; Thapa & Li, 2024), which either dynamically expand the model's capacity or isolate specific model weights to prevent interference between tasks; and (4) gradient-projection-based methods (Saha et al., 2021; Xiao et al., 2024), which project the current gradient update into the subspace defined by previous tasks to reduce forgetting.

### 2.2    CONTINUAL LEARNING FOR LLM

Continual learning for LLM can be categorized into: (1) subspace-based method (Wang et al., 2023a); (2) prompt-based method (Qin & Joty, 2022; Razdaibiedina et al., 2023); (3) attention-based method (Zhao et al., 2024); (4) architecture-based method (Wang et al., 2023b; 2024a). Orthogonal to these existing CL methods for LLMs, which rely on statistical correlations, these approaches (1) tend to memorize spurious features during new task learning, and (2) are highly vulnerable to data distribution shifts, exacerbating forgetting, particularly in limited data settings. In contrast, our method introduces a causality-based approach, which mitigates the impact of data distribution shifts on model parameters and lessens the memorization of spurious features during model training. This offers a novel strategy for improving continual learning performance in LLMs, ensuring better retention of old knowledge and generalization on new task.

## 3    CAUSAL MACHINE LEARNING

Causal inference (Pearl et al., 2016; Pearl & Mackenzie, 2018) focuses on uncovering cause-effect relationships among variables, providing a robust foundation for understanding and modeling data beyond simple correlations. The principles of causality have been increasingly integrated into machine learning (Parascandolo et al., 2018; Besserve et al., 2020), enabling models to incorporate causal reasoning for improved generalization and robustness. Causal machine learning methods have been applied to various domains, including few-shot learning (Yue et al., 2020), imitation learning (De Haan et al., 2019), domain adaptation (Gong et al., 2016; Magliacane et al., 2018; Kong et al., 2022) and representation learning (Schölkopf et al., 2021). These methods primarily leverage hard interventions, where variables are explicitly manipulated or fixed to isolate causal effects. While effective in many contexts, such approaches are not directly suitable for the continual learning framework. In continual learning, the ability to adapt to new tasks while retaining knowledge from prior tasks is critical, but hard interventions often result in static model parameters. This eliminates catastrophic forgetting but compromises the model's capacity to learn new tasks effectively.

In contrast, our proposed method introduces soft interventions, which adjust the dependency of model parameters on input distributions without freezing them entirely. This approach strikes a balance between mitigating forgetting and maintaining adaptability to new tasks. By dynamically refining the influence of input distributions during learning, our method ensures continual improvement across tasks while preserving past knowledge. To the best of our knowledge, this is the first application of causal principles via soft interventions in the context of continual learning, addressing a unique set of challenges not tackled by prior causal ML methods.

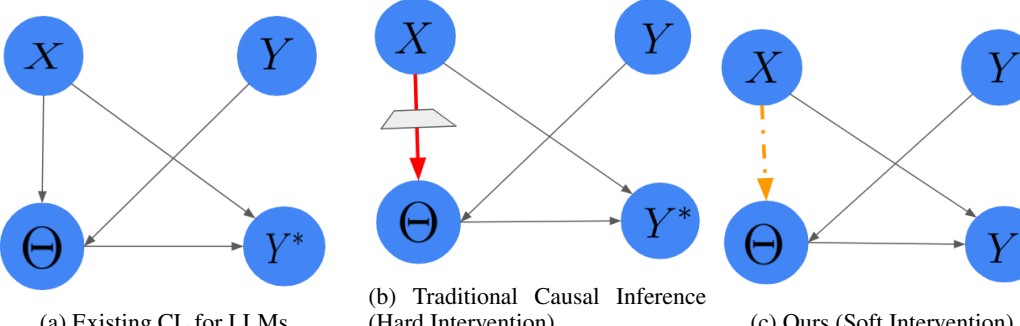

(a) Existing CL for LLMs     (b) Traditional Causal Inference (Hard Intervention)     (c) Ours (Soft Intervention)

Figure 1: Causal Diagram of CL in LLMs: the nodes represent key random variables: $X$ denotes the input data distribution, $Y$ denotes the ground truth data label, $Y^*$ denotes the model outputs and $\Theta$ represents the model parameters. The arrows $\rightarrow$ indicate causal relationships between these variables. Specifically, the causal link $X \rightarrow \Theta \leftarrow Y$ highlights two major challenges in CL: (i) Forgetting of Previously Learned Knowledge: As the data distribution $(X, Y)$ shifts over time, it directly influences the model parameters $\Theta$, causing them to change and resulting in the loss of previously acquired knowledge. (ii) Memorization of Spurious Features in New Tasks: Due to the strong dependence of $\Theta$ on $X$, the model tends to memorize spurious correlations in the data, leading to suboptimal performance on new tasks. (a) Traditional CL Methods: These methods do not employ any intervention to address the causal link between $X$ and $\Theta$. As a result, they suffer from the above challenges, including catastrophic forgetting and spurious feature memorization. (b) Hard Interventions in Traditional Causal Inference: Traditional causal inference methods propose hard interventions on the link $X \rightarrow \Theta$, effectively removing the influence of $X$ on $\Theta$. While this prevents the model from forgetting old knowledge, it also hampers the ability to learn new task-specific information, making the model static and unresponsive to new data. (c) Our Proposed Soft Intervention: We introduce a novel soft intervention approach that *partially* mitigates the influence of $X$ on $\Theta$, rather than completely severing the connection. This technique offers two key advantages: (1) Reduced forgetting: By lessening the impact of data distribution shifts on model parameters, the model retains previously learned knowledge more effectively while still incorporating new information. (2) Mitigated memorization of spurious features: The reduced reliance of $\Theta$ on $X$ makes the model less prone to overfitting spurious correlations, leading to better performance on new tasks.

## 4 DATA AUGMENTATION AND PARAMETER PERTURBATION

Traditional methods such as dropout, MixOut (Lee et al., 2020), and data augmentation primarily focus on improving generalization during model training by introducing noise to the input data or model parameters to prevent overfitting. However, these approaches overlook the dependency between model parameters and input data, which can result in suboptimal solutions. In contrast, our proposed method specifically targets mitigating the dependency of model parameters on input distributions, which is a crucial challenge in continual learning where input distributions change over time. This focus on reducing distribution dependency is not a primary design goal of existing methods like dropout or MixOut.

## 5 METHOD

In this section, we begin by outlining the problem setup and the limitations of existing CL approaches for LLMs in Section 5.1. Next, we introduce our proposed causality-guided CL approach in Section 5.2.

### 5.1 PROBLEM SETUP AND LIMITATIONS OF EXISTING WORKS

**Problem Setup and Formulation** Given a pre-trained LLM with parameters $\theta_0$, the goal of CL is to train the model sequentially on a series of $N$ tasks, each with its own training dataset $\mathcal{D}_1^{tr}, \mathcal{D}_2^{tr}, \ldots, \mathcal{D}_N^{tr}$. After training on the final task, the objective is to obtain a model $f_{\boldsymbol{\theta}}$ parameter-

ized by $\boldsymbol{\theta}$ that not only performs well on the current task but also retains high performance across the test sets of all previously learned tasks, i.e., $\mathcal{D}_1^{te}, \mathcal{D}_2^{te}, \ldots, \mathcal{D}_N^{te}$.

To mitigate forgetting during CL in LLMs, existing approaches aim to minimize the following loss function to update LLMs:

$$\mathcal{L} = \mathbb{E}_{(\boldsymbol{x}, y) \sim \mathcal{D}_t^{tr}}(\boldsymbol{x}, y, \boldsymbol{\theta}) + \lambda \mathcal{L}_{old}(\boldsymbol{\theta}) \tag{1}$$

where $\mathbb{E}_{(\boldsymbol{x}, y) \sim \mathcal{D}_t^{tr}}(\boldsymbol{x}, y, \boldsymbol{\theta})$ represents the loss on the training data of the current new task $t$, i.e., $\mathcal{D}_t^{tr}$. $\mathcal{L}_{old}(\boldsymbol{\theta})$ is the loss term designed to preserve previously learned knowledge and mitigate forgetting. This term could involve a regularization penalty on the model parameters or a memory replay loss. The hyperparameter $\lambda$ controls the balance between optimizing performance on the new task and preserving knowledge from old tasks. To better illustrate the limitations of CL optimization described in Eq. 1, we construct a causal diagram (Pearl, 2009) as explained below.

**Understanding CL in LLMs through a Causal Diagram**  Figure 1 illustrates the causal relationships involved in CL for LLMs using a causal diagram. In this diagram, each node represents a random variable: $X$ denotes the input data, $Y^*$ represents the model's prediction, $Y$ represents the data ground truth label, and $\boldsymbol{\Theta}$ symbolizes the model parameters. The arrows between nodes capture the causal dependencies between these variables. For instance, the arrows $X \to Y^* \leftarrow \boldsymbol{\Theta}$ suggest that both the input data ($X$) and the model parameters ($\boldsymbol{\Theta}$) influence the predictions ($Y^*$). Similarly, the arrow $X \to \boldsymbol{\Theta}$ indicates that shifts in the data distribution ($X$) directly affect the model parameters ($\boldsymbol{\Theta}$). These causal relationships highlight two key limitations in existing CL methods: (1) Changes in the data distribution ($X$) lead to shifts in the model parameters ($\boldsymbol{\Theta}$), which causes the model to forget previously learned knowledge as it adapts to new data. This phenomenon is known as catastrophic forgetting. (2) The model tends to memorize spurious features of the new task due to its heavy reliance on the input data ($X$), rather than identifying true causal relationships. Consequently, the performance on new tasks can degrade, especially when faced with unseen data. Figure 1a shows how existing CL approaches, which rely on optimizing Eq. 1, are fundamentally statistic-correlation-based. In these approaches, the model parameters $\boldsymbol{\Theta}$ are heavily influenced by the observed data statistics in $X$, making them vulnerable to shifts and spurious patterns in the data.

## 5.2 Causality-Guided Continual Learning for LLMs

To address the problem of existing approaches that heavily rely on the statistical correlation between input data $X$ and model parameters $\boldsymbol{\Theta}$ (i.e., $X \to \boldsymbol{\Theta}$ ), we propose a novel method grounded in causal inference. Our approach mitigates the causal dependency between the input variable $X$ and the model parameters $\boldsymbol{\Theta}$, enhancing the model's generalization.

**Traditional Causal Inference (Hard Intervention)**  The do-operation ($\text{do}(\boldsymbol{\Theta} = \boldsymbol{\theta})$): The do-operator, introduced by Judea Pearl (Pearl, 2009), is a key concept in causal inference, providing a formal framework to distinguish causation from mere correlation by modeling the effects of interventions in a system. The do-operation represents an intervention where a variable, in this case, $\boldsymbol{\Theta}$ (model parameters), is forcibly set to a specific value $\boldsymbol{\theta}$, disregarding its natural causes. This intervention allows us to estimate the direct causal effect of $\boldsymbol{\Theta}$ on another variable, such as $Y$ (model outputs), which is expressed as $P(y|\text{do}(\boldsymbol{\Theta} = \boldsymbol{\theta}))$. Unlike standard conditional probabilities like $P(y|\boldsymbol{\Theta} = \boldsymbol{\theta})$, which capture the observed relationships between $\boldsymbol{\Theta}$ and $Y$ in the data, $P(y|\text{do}(\boldsymbol{\Theta} = \boldsymbol{\theta}))$ models the outcome when $\boldsymbol{\Theta}$ is actively set to $\boldsymbol{\theta}$, effectively severing any natural influences on $\boldsymbol{\Theta}$. However, as illustrated in Figure 1b, implementing a hard intervention that completely severs the causal link between the input data $X$ and model parameters $\boldsymbol{\Theta}$ would result in the model being immune to forgetting because $\boldsymbol{\Theta}$ no longer depends on $X$. While this prevents catastrophic forgetting, it also hinders the model's ability to learn new tasks, as the model parameters remain fixed and unresponsive to new input data. This limitation emphasizes the trade-off inherent in traditional causal interventions within CL settings.

**Our Method (Soft Intervention)**  To overcome the limitations of hard interventions in traditional causal inference, we propose a novel approach: a partial (soft) intervention on the causal relationship between the data input $X$ and the model parameters $\boldsymbol{\Theta}$. Unlike hard interventions that completely disconnect $\boldsymbol{\Theta}$ from $X$, soft interventions allow for a controlled, partial dependence, effectively balancing the model's need to retain previously learned knowledge while still adapting to

new information. By softly intervening on $\boldsymbol{\Theta}$, we can reduce the influence of data shifts on model parameters, thereby mitigating catastrophic forgetting of previously learned tasks. At the same time, this approach retains enough flexibility to allow the model to learn new tasks effectively. The goal is to limit the degree to which $\boldsymbol{\Theta}$ is affected by changes in $X$, thereby reducing the impact of spurious correlations without entirely severing the ability of the model to adjust and learn. As illustrated in Figure 1c, our soft intervention approach modifies the influence of input data on model parameters, striking a balance that retains learned knowledge while enabling the integration of new information. This targeted, controlled intervention can be mathematically expressed as follows:

$$P(y|\text{do(soft}(\boldsymbol{\Theta}))) = \sum_{\boldsymbol{x}\in\mathcal{X}} P(\boldsymbol{x}) \sum_{\boldsymbol{\theta}\sim\boldsymbol{\Theta}} P(y|\boldsymbol{x},\boldsymbol{\theta})P'(\boldsymbol{\theta}|\boldsymbol{x}) \tag{2}$$

where $P'(\boldsymbol{\theta}|\boldsymbol{x})$ denotes the modified parameter distribution conditioned on input $\boldsymbol{x}$, reflecting the partial (soft) intervention on $\boldsymbol{\Theta}$. We set it to be $P'(\boldsymbol{\theta}|\boldsymbol{x}) \approx P(\boldsymbol{\theta}|\boldsymbol{x}) + \mathcal{N}(\mathbf{0}, \boldsymbol{\sigma}^2)$, where $\boldsymbol{\sigma}$ are learnable standard deviation parameters. This modified posterior distribution $P'(\boldsymbol{\theta}|\boldsymbol{x})$, diverges from the original $P(\boldsymbol{\theta}|\boldsymbol{x})$, enabling $\boldsymbol{\Theta}$ to partially depend on $X$. As a result, it reduces the strong dependency of $\boldsymbol{\Theta}$ on $X$. This equation is derived using backdoor adjustment from causal inference. The backdoor path is $\Theta \leftarrow X \rightarrow Y^*$, and the variable $X$ is not a descendant of $\Theta$. Therefore, $X$ satisfies the backdoor criterion with respect to the causal effect of $\Theta$ on $Y^*$. Consequently, we can apply the backdoor adjustment to account for the confounding variable and estimate the causal effect.

To reduce the reliance of the model parameters $\boldsymbol{\Theta}$ on data inputs $X$ and enhance the relation between $\boldsymbol{\Theta}$ and data label $Y$ in a soft way, we propose maximizing the following learning objective:

$$\mathcal{L}_{causal} = I(\boldsymbol{\Theta}, Y) - \alpha I(\boldsymbol{\Theta}, X) + \lambda \mathcal{L}_{old}(\boldsymbol{\theta}) \tag{3}$$

where $I(\boldsymbol{\Theta}, Y)$ denotes the mutual information between $\boldsymbol{\Theta}$ and $Y$, defined as $I(\boldsymbol{\Theta}, Y) = \int P(y, \boldsymbol{\theta}) \log \frac{P(y,\boldsymbol{\theta})}{P(y)P(\boldsymbol{\theta})} dy d\boldsymbol{\theta}$, which quantifies the dependence or shared information between the variables $Y$ and $\boldsymbol{\Theta}$. Maximizing $I(\boldsymbol{\Theta}, Y)$ would enhance the prediction of label $y$ with parameters $\boldsymbol{\theta}$. $I(\boldsymbol{\Theta}, X)$ denotes the mutual information between $\boldsymbol{\Theta}$ and $X$. Minimizing $I(\boldsymbol{\Theta}, X)$ reduces the dependence of $\boldsymbol{\Theta}$ on $X$. $\alpha > 0$ is a weighting constant that balances the two mutual information terms. However, calculating mutual information is computationally intractable due to the difficulty of modeling the joint distribution $P(y, \boldsymbol{\theta})$, and marginal distribution $P(\boldsymbol{\theta})$ in high-dimensional spaces. Therefore, inspired by the variational inference Alemi et al. (2017), we propose maximizing the following variational objective:

$$\mathcal{L}_{causal} \approx \frac{1}{M} \sum_{i=1}^{i=M} \mathbb{E}_{\boldsymbol{\epsilon}\sim P(\boldsymbol{\epsilon})}[\log P(y|\text{do(soft}(\boldsymbol{\Theta})), \boldsymbol{\epsilon})] - \alpha \mathbb{KL}(P'(\boldsymbol{\theta}|\boldsymbol{x}_i), Q(\boldsymbol{\theta})) + \lambda \mathcal{L}_{old}(\boldsymbol{\theta}) \tag{4}$$

where $M$ denotes the number of training data points. $P'(\boldsymbol{\theta}|\boldsymbol{x}_i)$ denotes the modified model parameter posterior distribution given the input $\boldsymbol{x}_i$ as Eq. 2, and $Q(\boldsymbol{\theta})$ is the model prior distribution, which is set to be standard normal distribution, i.e., $Q(\boldsymbol{\theta}) = \mathcal{N}(0, I)$. We put the detailed derivations of Eq. 4 in Appendix A. In addition, to simulate samples from the distribution $P(\boldsymbol{\epsilon})$, i.e., generate noisy versions of the input text, we can use the following transformations: (1) WordNet synonym replacement, which randomly replaces words with their synonyms; (2) Word deletion, which randomly removes words from the sentence; (3) Word order swaps, which randomly swap the positions of words in the sentence; and (4) Random synonym insertion, which inserts a synonym of a random word at a random position. These techniques collectively transform the input text to build noisy data from $P(\boldsymbol{\epsilon})$. Furthermore, it is important to note that the last term in Eq. 4, $\lambda \mathcal{L}_{old}(\boldsymbol{\theta})$, represents the loss term introduced in previous works to mitigate forgetting of old knowledge. The hyperparameter $\lambda$ is thus not part of our method. Consequently, $\alpha$ is the only hyperparameter specific to our approach. We refer to our proposed approach as Causality-Guided Continual Learning (**CGCL**). The detailed steps of our algorithm are provided in Algorithm 1.

---

**Algorithm 1** Causality-Guided Continual Learning for LLMs.

---

1: **REQUIRE:** pre-trained LLM parameters $\boldsymbol{\theta}_0$, learning rate $\eta$.
2: **for** $n = 1$ to $N$ **do** (number of CL tasks)
3:     **for** $k = 1$ to $K$ **do** (number of CL steps)
4:         calculate the data intervention by Eq. 2.
5:         calculate the causal learning objective by Eq. 4.
6:         update the LLM parameters by $\boldsymbol{\theta}_{k+1}^n = \boldsymbol{\theta}_k^n - \eta\nabla\mathcal{L}_{causal}(\boldsymbol{\theta}_k^n)$
7:     **end for**
8: **end for**

---

## 6 EXPERIMENT

In this Section, we first provide the experiment setup in Section 6.1. Then, we present the experiment results in Section 6.2. Next, we present more detailed analysis and ablation study in Section 6.3.

### 6.1 SETUP

**Datasets** Following (Razdaibiedina et al., 2023; Wang et al., 2023a), we use the benchmark that includes a variety of NLP tasks, each accompanied by expert-crafted instructions, to provide a more practical evaluation framework for CL in LLMs. We use two benchmark datasets to evaluate the performance of CL methods in LLMs.

- *standard CL benchmark*: The four text classification tasks (Zhang et al., 2015) are shuffled into three distinct sequences, forming order 1, 2, and 3, for use in the standard CL benchmark.

- *long sequence CL benchmark*: This benchmark consists of a total of 15 tasks, which present additional challenges to existing CL approaches. Specifically, it includes five tasks from the CL benchmark, four tasks from the GLUE benchmark (MNLI, QQP, RTE, SST2) (Wang et al., 2019b), five tasks from the SuperGLUE benchmark (WiC, CB, COPA, MultiRC, BoolQ) (Wang et al., 2019a), and the IMDB movie reviews dataset (Maas et al., 2011), creating long sequence CL benchmark orders 4, 5, and 6. The task sequence order can be found in Appendix B.

**Baselines** We compare to the following SOTA baselines: LFPT5 (Qin & Joty, 2022) is a replay-based method that continuously trains a soft prompt designed to both solve current tasks and generate pseudo-labeled samples from previously learned domains. These generated samples are then utilized in experience replay to reinforce past knowledge. EPI (Wang et al., 2023b) employs a parameter isolation strategy to reduce forgetting by allocating a small set of task-specific parameters for each task, which are learned alongside a shared pre-trained model. ProgPrompt (Razdaibiedina et al., 2023) incrementally learns a new soft prompt for each new task and sequentially append it to the prompts learned from previous tasks. O-LoRA (Wang et al., 2023a) which builds on LoRA (Hu et al., 2022) and learns tasks in distinct low-rank vector subspace that are maintained orthogonal to each other, effectively minimizing interference between tasks. SAPT (Zhao et al., 2024) aligns the parameter-efficient tuning (PET) learning and selection through a shared attention-based learning and selection module. In addition, we also compare to the data augmentation (DA) techniques in (Wei & Zou, 2019) as an additional strong baseline.

**Pre-trained Models** Following the setting in (Wang et al., 2023a; Zhao et al., 2024), we use the pre-trained T5-Large (Raffel et al., 2020) and LLaMA-2-7B (Touvron et al., 2023).

**Evaluation Metrics** We define $a_{i,j}$ as the evaluation performance on the $j$-th task after training on the $i$-th task. Following (Wang et al., 2023a), we utilize the average performance across the CL task sequence to assess the performance of CL on LLM. The overall performance across all tasks after completing the training on the final task is defined as: $A_\mathcal{T} = \frac{1}{\mathcal{T}}\sum_{t=1}^{\mathcal{T}} a_{\mathcal{T},t}$.

**Implementation Details** Our implementation is based on the codebase of O-LORA (Wang et al., 2023a). All the experiment results are averaged over 3 runs and are performed on $4\times$A6000 NVIDIA

Table 1: The overall results on two continual learning benchmarks with **T5-Large** model with **sample size of 100**.

| | Standard CL Benchmark | | | | Long Sequence Benchmark | | | |
|---|---|---|---|---|---|---|---|---|
| | Order-1 | Order-2 | Order-3 | Avg | Order-4 | Order-5 | Order-6 | Avg |
| ProgPrompt | 55.39 | 56.67 | 51.38 | 54.48 | 46.28 | 48.91 | 36.75 | 43.98 |
| LFPT5 | 52.20 | 54.31 | 49.52 | 52.01 | 40.46 | 46.25 | 32.83 | 39.85 |
| EPI | 43.62 | 47.29 | 42.09 | 44.33 | 31.43 | 40.37 | 27.51 | 33.10 |
| SAPT-LoRA | 60.51 | 61.77 | 58.23 | 60.17 | 47.34 | 55.89 | 40.93 | 48.05 |
| O-LoRA | 59.42 | 59.46 | 55.68 | 58.19 | 48.20 | 52.06 | 39.78 | 46.68 |
| O-LoRA + DA | 61.83 | 62.61 | 59.97 | 61.47 | 51.57 | 54.72 | 43.64 | 49.98 |
| O-LoRA + CGCL | **73.66±2.03** | **75.38±1.76** | **72.75±1.94** | **73.93±1.89** | **64.81±2.53** | **67.60±2.67** | **60.87±2.36** | **64.43±2.51** |

Table 2: The overall results on two continual learning benchmarks with **LLaMA-2-7B** model with **sample size of 100**.

| | Standard CL Benchmark | | | | Long Sequence Benchmark | | | |
|---|---|---|---|---|---|---|---|---|
| | Order-1 | Order-2 | Order-3 | Avg | Order-4 | Order-5 | Order-6 | Avg |
| ProgPrompt | 62.03 | 45.69 | 61.40 | 56.37 | 52.35 | 51.82 | 52.31 | 52.16 |
| LFPT5 | 57.17 | 42.51 | 53.76 | 51.15 | 46.08 | 47.34 | 50.75 | 48.06 |
| EPI | 46.32 | 32.49 | 52.71 | 43.84 | 43.24 | 42.53 | 43.67 | 43.15 |
| SAPT-LoRA | **69.37** | 51.95 | 67.83 | 63.05 | 59.63 | 59.21 | 61.32 | 60.05 |
| O-LoRA | 68.24 | 50.63 | 65.68 | 61.52 | 57.45 | 56.79 | 59.28 | 57.84 |
| O-LoRA + DA | 69.05 | 52.76 | 67.72 | 63.18 | 59.91 | 59.68 | 62.86 | 60.82 |
| O-LoRA + CGCL | 67.91±1.93 | **69.94±1.82** | **72.17±1.58** | **70.01±1.76** | **66.71±2.06** | **65.35±1.97** | **69.99±1.83** | **67.35±1.91** |

GPU using DeepSpeed repository. We set the word perturbation rate in each sentence to 10% in order to generate noisy samples from $P(\epsilon)$, and $\alpha = 0.003$. LoRA configuration: $r = 8$. The learning rate is set to be 1e-4. More implementation details can be found in Appendix C.

## 6.2 RESULTS

We present the CL results with a sample size of 100 per task for the T5-Large model in Table 1 and the LLaMA-2-7B model in Table 2. Our method significantly exceeds the SOTA CL performance, achieving improvements of over 12% with T5-Large and more than 7% with LLaMA-2-7B on the standard CL benchmark. On the long-sequence CL benchmark, our approach demonstrates even greater gains, improving SOTA performance by over 15% with T5-Large and more than 6% with LLaMA-2-7B. These results emphasize the effectiveness of our causality-guided approach in data-efficient learning scenarios.

**Effect of Sample Size for Each Task** To evaluate the impact of sample size on task performance, we conduct experiments using a sample size of 500 for each task. This larger sample size allows us to explore how varying the amount of training data influences the effectiveness of our CGCL approach. The CL results for the T5-large model with a sample size of 500 are presented in Table 3, while the results for the LLaMA-2-7B model are detailed in Table 4. Our findings indicate that even with this increased sample size, our method continues to deliver substantial improvements in overall CL performance. This suggests that our approach is robust and effective, demonstrating its ability to enhance learning outcomes regardless of the amount of available training data. The consistent performance gains across both models further validate the advantages of incorporating a causality-guided perspective in CL tasks.

Table 3: The overall results on two continual learning benchmarks with **T5-Large** model with **sample size of 500**.

| | Standard CL Benchmark | | | | Long Sequence Benchmark | | | |
|---|---|---|---|---|---|---|---|---|
| | Order-1 | Order-2 | Order-3 | Avg | Order-4 | Order-5 | Order-6 | Avg |
| ProgPrompt | 70.31 | 71.18 | 72.05 | 71.18 | 52.20 | 64.31 | 62.16 | 59.56 |
| LFPT5 | 62.04 | 61.37 | 60.90 | 61.43 | 47.95 | 46.26 | 43.35 | 45.85 |
| EPI | 64.29 | 66.73 | 65.64 | 65.55 | 49.32 | 61.43 | 58.62 | 56.46 |
| SAPT-LoRA | 75.32 | 75.86 | 76.37 | 75.85 | 60.28 | 70.24 | 69.83 | 66.78 |
| O-LoRA | 74.15 | 75.29 | 75.71 | 75.05 | 59.83 | 69.38 | 69.20 | 66.14 |
| O-LoRA + DA | **75.81** | 76.52 | 76.93 | 76.42 | 62.56 | 70.91 | 71.22 | 68.23 |
| O-LoRA + CGCL | 75.39±1.65 | **78.16±1.72** | **78.42±1.43** | **77.32±1.59** | **69.30±1.90** | **72.39±1.87** | **76.88±1.16** | **72.86±1.68** |

Table 4: The overall results on two continual learning benchmarks with **LLaMA-2-7B** model with **sample size of 500**.

| | Standard CL Benchmark | | | | Long Sequence Benchmark | | | |
|---|---|---|---|---|---|---|---|---|
| | Order-1 | Order-2 | Order-3 | Avg | Order-4 | Order-5 | Order-6 | Avg |
| ProgPrompt | 62.56 | 70.25 | 60.58 | 64.46 | 57.03 | 56.67 | 61.82 | 58.51 |
| LFPT5 | 51.43 | 60.18 | 54.71 | 55.44 | 55.97 | 50.60 | 52.29 | 52.95 |
| EPI | 57.95 | 53.52 | 59.41 | 56.96 | 53.35 | 52.38 | 54.67 | 53.47 |
| SAPT-LoRA | 68.70 | 75.67 | 70.32 | 71.56 | 64.51 | 63.42 | 66.35 | 64.76 |
| O-LoRA | 66.51 | 73.36 | 68.23 | 69.37 | 62.64 | 61.06 | 65.42 | 63.04 |
| O-LoRA + DA | 68.62 | 75.79 | 71.15 | 71.85 | 63.89 | 62.90 | 66.78 | 64.52 |
| O-LoRA + CGCL | **74.81±1.08** | **78.03±0.95** | **79.61±0.79** | **77.48±0.93** | **68.36±1.02** | **66.54±1.29** | **69.46±1.30** | **68.12±1.21** |

## 6.3 ANALYSIS

**Individual Task Performance Evaluation** To assess the performance of individual (both old and new) tasks, we conduct a comparative analysis of each task under a data limit of 100 examples, utilizing both the T5-large model and LLaMA-2-7B within the context of task order 6. The performance results for the T5-large model are presented in Figure 2, while the corresponding results for LLaMA-2-7B are depicted in Figure 3. Our findings reveal that, with the implementation of our CGCL approach, the performance of each task shows a significant improvement across most datasets. This enhancement can be attributed to the effectiveness of our causality-guided methodology, which diminishes the dependency of model parameters on input data. Specifically, this reduction in dependency leads to two key benefits: (1) it mitigates the risk of forgetting previously learned task knowledge, and (2) it lessens the likelihood of memorizing spurious features associated with new tasks. Consequently, our proposed approach elevates superior overall performance across both previously learned old and new tasks.

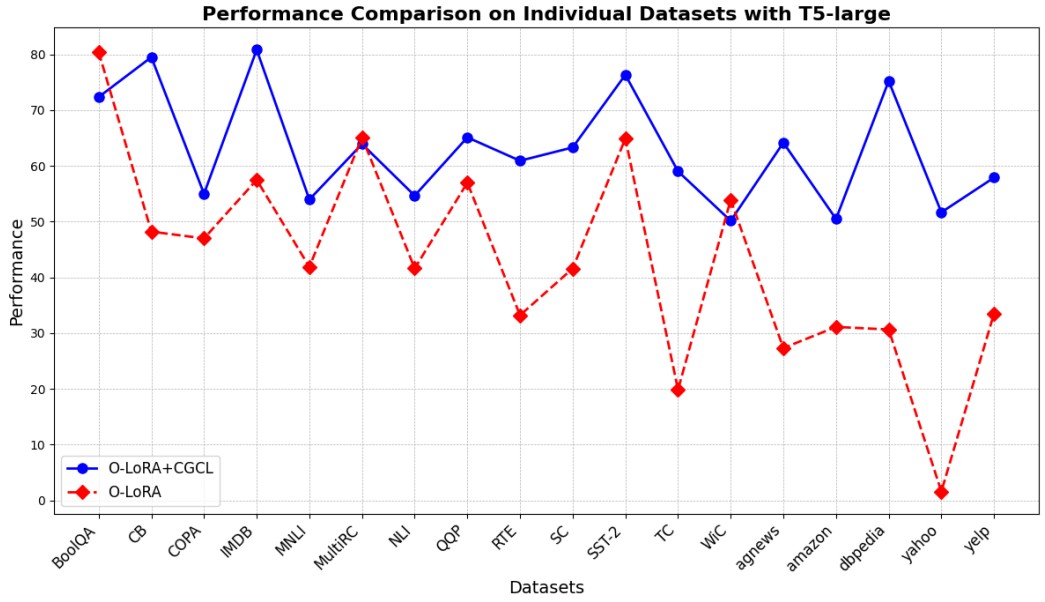

Figure 2: Performance of individual tasks in the CL sequence using the T5-large model, with a data limit of 100 samples per task under the task order 6.

**Hyperparameter Sensitivity Analysis** To evaluate the sensitivity of the hyperparameter $\alpha$ in Eq. 4, we perform a sensitivity analysis, with the results summarized in Table 5. The findings indicate that our method is not very sensitive to $\alpha$.

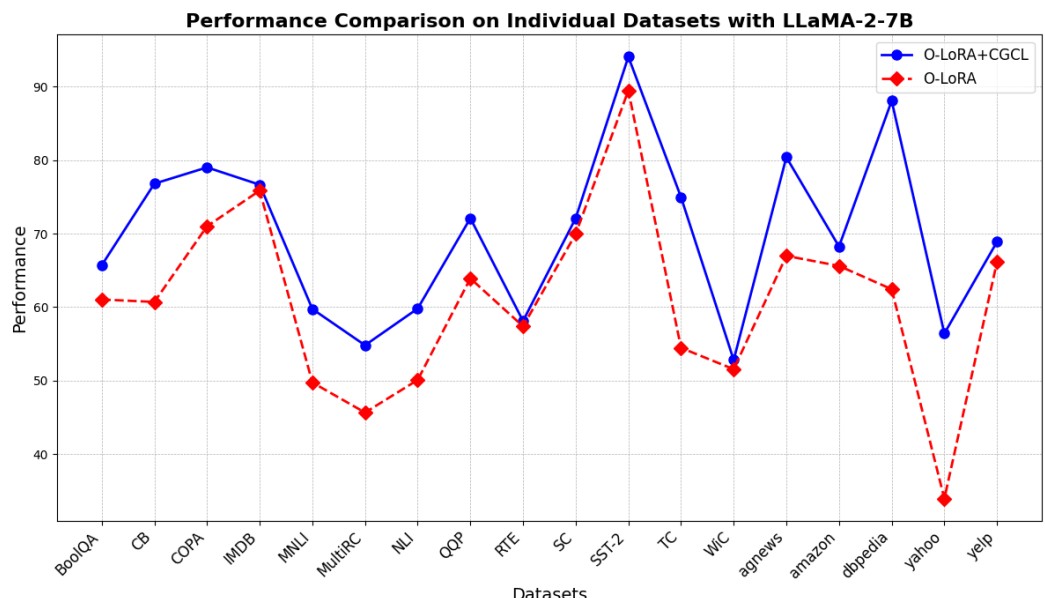

Figure 3: Performance of individual tasks in the CL sequence using the LLaMA-2-7B model, with a data limit of 100 samples per task under the task order 6.

Table 5: Hyperparameter analysis of $\alpha$ on CL performance under sample size of 100 with T5-large

| $\alpha$ | Order 4 | Order 5 | Order 6 |
|---|---|---|---|
| 0.001 | 62.73 | 68.17 | 62.32 |
| 0.003 | 64.81 | 67.60 | 60.87 |
| 0.005 | 63.06 | 65.28 | 61.53 |

## 7 CONCLUSION

This paper explores continual learning for LLMs in a limited data setting. We introduce a novel causality-guided approach that addresses two key challenges: mitigating forgetting of previously learned tasks and avoiding the memorization of spurious features in new task data. Extensive experiments on large-scale pre-trained models, including T5-large and LLaMA-2, highlight the effectiveness of the proposed method.

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

**Appendix**

## A  Causal Learning Objective

$$\mathcal{L}_{causal} = I(\boldsymbol{\Theta}, Y) - \alpha I(\boldsymbol{\Theta}, X) \tag{5}$$

To minimize Eq. 5, inspired by (Alemi et al., 2017), we obtain the following the variational objective:

Given $I(\boldsymbol{\Theta}, Y)$ and $I(\boldsymbol{\Theta}, X)$, we have the following derivation.

(1) For $I(\boldsymbol{\Theta}, Y)$, we can derive the following inequality:

$$I(\boldsymbol{\Theta}, Y) = \int d\boldsymbol{\theta} \, dy \, P(\boldsymbol{x}, \boldsymbol{\theta}) \log \frac{P(\boldsymbol{\theta}, y)}{P(\boldsymbol{\theta})P(y)} = \int d\boldsymbol{\theta} \, dy \, P(y, \boldsymbol{\theta}) \log \frac{P(y|\boldsymbol{\theta})}{P(y)} \tag{6}$$

For $\mathbb{KL}(p(Y|\boldsymbol{\Theta}), q(Y|\boldsymbol{\Theta})) \geq 0$, we can have the following inequality:

$$\int dy d\boldsymbol{\theta} P(y|\boldsymbol{\theta}) \log P(y|\boldsymbol{\theta}) \geq \int dy d\boldsymbol{\theta} P(y|\boldsymbol{\theta}) \log q(y|\boldsymbol{\theta}) \tag{7}$$

Then, we have the following inequality:

$$I(\boldsymbol{\Theta}, Y) \geq \int d\boldsymbol{\theta} \, dy \, P(y, \boldsymbol{\theta}) \log \frac{q(y|\boldsymbol{\theta})}{P(y)} \tag{8}$$

$$= \int d\boldsymbol{\theta} \, dy \, P(y, \boldsymbol{\theta}) \log q(y|\boldsymbol{\theta}) - \int dy \, P(y) \log P(y) \tag{9}$$

Since $P(y) \log P(y)$ does not depend on the model parameters $\boldsymbol{\theta}$, so this term could be ignored.

We can also obtain the following equality:

$$P(y, \boldsymbol{\theta}) = \int d\boldsymbol{x} P(\boldsymbol{x}, y, \boldsymbol{\theta}) = \int d\boldsymbol{x} P(\boldsymbol{x}) P(y|\boldsymbol{x}) P(\boldsymbol{\theta}|\boldsymbol{x}) \tag{10}$$

By plugging-in Eq. (10) into Eq. (8), we can have the following bound:

$$I(\boldsymbol{\Theta}, Y) \geq \int d\boldsymbol{x} \, dy \, d\boldsymbol{\theta} \, P(\boldsymbol{x}) P(y|\boldsymbol{x}) P(\boldsymbol{\theta}|\boldsymbol{x}) \log q(y|\boldsymbol{\theta}). \tag{11}$$

(2) For $I(\boldsymbol{\Theta}, X)$, we can derive the following inequality:

$$I(\boldsymbol{\Theta}, X) = \int d\boldsymbol{\theta} \, d\boldsymbol{x} \, P(\boldsymbol{x}, \boldsymbol{\theta}) \log \frac{P(\boldsymbol{\theta}, \boldsymbol{x})}{P(\boldsymbol{\theta})P(\boldsymbol{x})} = \int d\boldsymbol{\theta} \, d\boldsymbol{x} \, P(\boldsymbol{x}, \boldsymbol{\theta}) \log \frac{P(\boldsymbol{\theta}|\boldsymbol{x})}{P(\boldsymbol{\theta})} \tag{12}$$

$$= \int d\boldsymbol{\theta} \, d\boldsymbol{x} \, P(\boldsymbol{x}, \boldsymbol{\theta}) \log P(\boldsymbol{\theta}|\boldsymbol{x}) - \int d\boldsymbol{\theta} \, P(\boldsymbol{\theta}) \log P(\boldsymbol{\theta}) \tag{13}$$

where

$$P(\boldsymbol{\theta}) = \int d\boldsymbol{x} \, P(\boldsymbol{\theta}|\boldsymbol{x}) \, P(\boldsymbol{x}) \tag{14}$$

We can use $r(\boldsymbol{\theta})$ be a variational approximation to $P(\boldsymbol{\theta})$, according to the inequality of $\mathbb{KL}(P(\boldsymbol{\theta}), r(\boldsymbol{\theta})) \geq 0$, we can obtain the following inequality:

$$\int d\boldsymbol{\theta} P(\boldsymbol{\theta}) \log P(\boldsymbol{\theta}) \geq \int d\boldsymbol{\theta} P(\boldsymbol{\theta}) \log r(\boldsymbol{\theta}) \tag{15}$$

Based on Eq. (14), Eq. (12) can be rewritten as (16)

$$I(\boldsymbol{\Theta}, X) \leq \int d\boldsymbol{x} \, d\boldsymbol{\theta} \, P(\boldsymbol{x}) P(\boldsymbol{\theta}|\boldsymbol{x}) \log \frac{P(\boldsymbol{\theta}|\boldsymbol{x})}{Q(\boldsymbol{\theta})}. \tag{16}$$

Combining the two inequality (11) and (16), we have that:

$$I(\boldsymbol{\Theta}, Y) - I(\boldsymbol{\Theta}, X) \geq \int d\boldsymbol{x} dy d\boldsymbol{\theta} \, P(\boldsymbol{x}) P(y|\boldsymbol{x}) P(\boldsymbol{\theta}|\boldsymbol{x}) \log q(y|\boldsymbol{\theta}) \tag{17}$$

$$- \int d\boldsymbol{x} d\boldsymbol{\theta} \, P(\boldsymbol{x}) P(\boldsymbol{\theta}|\boldsymbol{x}) \log \frac{P(\boldsymbol{\theta}|\boldsymbol{x})}{Q(\boldsymbol{\theta})} = L \tag{18}$$

To efficiently calculate Eq. (17), we approximate the $P(\boldsymbol{x}, \boldsymbol{y}) = P(\boldsymbol{x}) P(\boldsymbol{y}|\boldsymbol{x})$ by empirical data distribution $P(\boldsymbol{x}, \boldsymbol{y}) = \frac{1}{N} \sum_{n=1}^{N} \delta_{\boldsymbol{x}_n}(\boldsymbol{x}) \delta_{\boldsymbol{y}_n}(\boldsymbol{y})$, $\delta_{\boldsymbol{x}_n}$ is the Dirac delta function on $\boldsymbol{x}_n$ and $\delta_{\boldsymbol{y}_n}$ is the Dirac delta function on $\boldsymbol{y}_n$.

Then we can use the reparameterization trick (Kingma & Welling, 2013) to rewrite the $P(\boldsymbol{\theta}|\boldsymbol{x}) d\boldsymbol{\theta} = P(\boldsymbol{\epsilon}) d\boldsymbol{\epsilon}$. This enables the distribution $P(\boldsymbol{\theta}|\boldsymbol{x})$ to be reparameterized as a function of $\boldsymbol{\epsilon}$. We then calculate the KL divergence between $P(\boldsymbol{\theta}|\boldsymbol{x})$ and $Q(\boldsymbol{\theta})$, and combine all together to minimize the following empirical loss function:

$$\mathcal{L}_{causal} \approx \frac{1}{M} \sum_{i=1}^{i=M} \mathbb{E}_{\boldsymbol{\epsilon} \sim P(\boldsymbol{\epsilon})} [\log P(y_i|f(\boldsymbol{x}_i, \boldsymbol{\epsilon}))] - \alpha \mathbb{KL}(P(\boldsymbol{\theta}|\boldsymbol{x}_i), Q(\boldsymbol{\theta})) \tag{19}$$

Next, we derive the soft intervention formula as below:

$$P(y|\text{do}(\text{soft}(\boldsymbol{\Theta}))) = \sum_{\boldsymbol{x} \in X} \sum_{\boldsymbol{\theta} \sim \boldsymbol{\Theta}} P(y|\boldsymbol{x}, \boldsymbol{\theta}) P'(\boldsymbol{\theta}|\boldsymbol{x}) P(\boldsymbol{x}) \text{(by adjustment formula or backdoor adjustment)} \tag{20}$$

$$= \sum_{\boldsymbol{x} \in \mathcal{X}} P(\boldsymbol{x}) \sum_{\boldsymbol{\theta} \sim \boldsymbol{\Theta}} P(y|\boldsymbol{x}, \boldsymbol{\theta}) P'(\boldsymbol{\theta}|\boldsymbol{x}) \tag{21}$$

To achieve causal learning during CL, we plug-in the Eq. 21 into Eq. 19, we obtain the following loss function for LLMs:

$$\mathcal{L}_{causal} \approx \frac{1}{M} \sum_{i=1}^{i=M} \mathbb{E}_{\boldsymbol{\epsilon} \sim P(\boldsymbol{\epsilon})} [\log P(y|\text{do}(\text{soft}(\boldsymbol{\Theta})), \boldsymbol{\epsilon})] - \alpha \mathbb{KL}(P'(\boldsymbol{\theta}|\boldsymbol{x}_i), Q(\boldsymbol{\theta})) \tag{22}$$

## B  BENCHMARK AND TASK SEQUENCE DETAILS

| Benchmark | Order | Task Sequence |
|---|---|---|
| Standard CL | 1 | dbpedia → amazon → yahoo → ag |
| | 2 | dbpedia → amazon → ag → yahoo |
| | 3 | yahoo → amazon → ag → dbpedia |
| Long sequence | 4 | mnli → cb → wic → copa → qqp → boolqa → rte → imdb → Yelp → amazon → sst-2 → dbpedia → ag → multirc → yahoo |
| | 5 | multirc → boolqa → wic → mnli → cb → copa → qqp → rte → imdb → sst-2 → dbpedia → ag → Yelp → amazon → yahoo |
| | 6 | Yelp → amazon → mnli → cb → copa → qqp → rte → imdb → sst-2 → dbpedia → ag → yahoo → multirc → boolqa → wic |

Table 6: Task Sequence Orders for CL in LLMs Experiments. Orders 1-3 correspond to the traditional task sequences commonly used in standard continual learning benchmarks (Zhang et al., 2015). Orders 4-6 expand on this by introducing longer sequences, each comprising 15 tasks (Razdaibiedina et al., 2023).

## C  IMPLEMENTATION DETAILS

- For T5-large model, the per device train batch size is set to be 8, per device eval batch size is set to be 64 and gradient accumulation steps is set to be 4.

- For Llama2-7B model, the per device train batch size is set to be 4, per device eval batch size is set to be 10 and gradient accumulation steps is set to be 4.

