# OpenReview forum: "Data Efficient Continual Learning of Large Language Model"
_ICLR.cc/2025/Conference — Submitted to ICLR 2025_

### Official Review · Reviewer_r6CX · 2024-10-31

**Soundness:** 2
**Presentation:** 2
**Contribution:** 2
**Rating:** 3
**Confidence:** 5

**Summary:**

This paper presents a causality-guided approach for data-efficient continual learning in large language models (LLMs), targeting the challenges of catastrophic forgetting and reliance on spurious correlations when faced with limited data. By applying soft causal interventions to decouple model parameters from the input data distribution, the method aims to retain past knowledge and generalize better across tasks. Experimental results demonstrate that this approach significantly improves performance over state-of-the-art continual learning methods on benchmarks using pre-trained LLMs, including T5-large and LLaMA-2.

**Strengths:**

1. The paper introduces an innovative framework that addresses continual learning from a causal inference perspective, effectively reducing forgetting and over-reliance on spurious correlations.
2. By implementing a soft intervention approach, the method achieves a balance between retaining past knowledge and adapting to new tasks, enhancing both old and new task performance.
3. Extensive experiments across multiple benchmarks and model types validate the approach's efficacy, showing significant improvements over current continual learning techniques.

**Weaknesses:**

1 Insufficient Validation of the Role of α in Reducing I(Θ, X):
Although the sensitivity analysis indicates that the method is not highly dependent on variations in α, the lack of a comparison experiment with α set to zero fails to clearly demonstrate α’s impact on controlling I(Θ, X) and maintaining memory retention. Including an experiment with α set to zero would more directly illustrate α's specific role in the process.
2 Limited Evaluation Metrics and Baseline Selection:
The evaluation metrics used in the experiments are somewhat limited and lack specific definitions, which weakens the credibility of the proof of concept. Additionally, the selection of baselines omits the inclusion of strong intervention methods, such as "hard interventions," which are prominent in forgetting mitigation research. Including a more comprehensive set of baselines, particularly with "hard interventions," would better highlight the strengths and weaknesses of the proposed method.
3 Insufficient Analysis of Task Performance Variance and Forgetting Effects:
The reliance on average performance metrics may obscure task-specific performance fluctuations or forgetting effects. In the long sequence CL benchmark (e.g., Task Order 6), proving that the model has not forgotten initial tasks (such as Yelp) would require re-evaluating the task performance after each newly added task, rather than only comparing the final overall performance. This more detailed assessment approach could use metrics like Forgetting Rate or Stability-Plasticity Trade-off to more thoroughly evaluate the model’s stability in task performance.
4 Small Experiment Data Size May Lack Representativeness:
The experiments use only 100 or 500 samples per task, which may not sufficiently represent the complexity of practical applications. While the method performs well under low-resource conditions, its effectiveness on larger-scale data has not been tested. Testing with larger datasets (e.g., thousands or tens of thousands of samples) and comparing these results to the low-resource experiments would more comprehensively demonstrate the method’s stability and adaptability.

**Questions:**

1 Limitations of Experiment Data Volume:
The experiments in this study use only 100 or 500 samples per task, which may not adequately reflect the complexity of real-world applications. Since practical tasks typically involve larger datasets, the applicability of results from such a limited dataset size to real-world scenarios is uncertain. This constraint could impact the method's feasibility in broader applications.
2 Applicability to Large-Scale Tasks:
While the study focuses on performance in low-resource conditions, it has yet to demonstrate whether the method achieves comparable results on large-scale, real-world tasks. Testing on larger datasets would be essential to confirm the method’s robustness and effectiveness for practical, extensive applications.
3 Lack of Comparison with No Training and Large-Scale Training Data:
The paper does not provide results for the model's performance without training or with larger training datasets, which limits a full assessment of the method’s robustness in different data conditions. Adding evaluations on larger datasets would offer a clearer view of the method's stability and generalizability across varying data environments.

---

### Official Review · Reviewer_i2KP · 2024-11-04

**Soundness:** 3
**Presentation:** 3
**Contribution:** 2
**Rating:** 5
**Confidence:** 4

**Summary:**

The paper proposes a novel causality-guided approach for continual learning in large language models (LLMs) to mitigate catastrophic forgetting and reliance on spurious features. The method introduces a soft intervention mechanism to reduce the dependency of model parameters on input data and enhances generalization across tasks. Extensive experiments on T5-large and LLaMA-2 models demonstrate significant improvements over state-of-the-art continual learning methods, especially under limited data.

**Strengths:**

* The paper explores the connection between the learning of spurious features and catastrophic forgetting, offering a novel perspective on continual learning and
addresses a critical challenge in the field using the idea of causal inference.
* The experiments are comprehensive, including multiple benchmarks, task order and sample sizes. Comparison is performed with strong baselines.
* The paper is well-written, with clear explanations of the methodology and the experiments.

**Weaknesses:**

Appropriateness of the scope of the paper:
* The paper proposes a causality-guided learning algorithm for continual learning in LLMs, but the algorithm itself is neither directly related to continual learning nor to LLMs. The core idea of the algorithm is a general approach to reduce the dependency of model parameters on input distributions, which could be applied to a broad range of machine learning models and tasks. The contribution and the discussion of the paper could be positioned more clearly be recognizing the proposed approach as a general causal machine learning method.

Citation of causal ML literature:
* The proposed method belongs to the field of causal machine learning, but the paper does not sufficiently discuss the existing literature on causal ML or compare the proposed method with existing methods in causal ML, besides citing the Judea Pearl paper. This could make it hard for readers to appreciate the contribution of the paper in the field of causal ML.

Novelty of the proposed method:
* The proposed method has two essential components to reduce the dependency of model parameters on input distributions: noisy perturbation of model parameters (line 246-250) and noisy perturbation of input data (line 270-280). Both of these are quite common techniques in machine learning to reduce overfitting and improve generalization. For example, dropout and MixOut to perturb model parameters, and data augmentation (paraphrase, back-translation, etc.) to perturb input data. Such methods can be readily used in continual learning and LLMs to improve generalization. The paper should discuss how the proposed method is different from these existing methods and why the proposed method is more effective.

Analysis of the proposed method:
* Ablation study: the paper does not include an ablation study to analyze the effectiveness of each component of the proposed method. This would help understand the contribution of each component to the overall performance.
* Hyperparameter sensitivity analysis: the result shows that the proposed method is not very sensitive to the hyperparameter α, and good result seems to be obtained even with a small α. Would this mean that the KL term in Eq. (4) is not very useful? Also, how do you tradeoff between more learning and less forgetting if result is not sensitive to α?
* Analysis beyond final performance: the paper claims that the method "lessens the likelihood of memorizing spurious features" (line 418), how do you show that the memorization of spurious features is indeed reduced? For example, maybe it is possible to analyze the model's attention weights or the model's prediction on the training data to show that the model memorized less spurious features.

**Questions:**

Please refer to the above section.

---

### Official Review · Reviewer_WY6a · 2024-11-04

**Soundness:** 3
**Presentation:** 2
**Contribution:** 3
**Rating:** 6
**Confidence:** 3

**Summary:**

This paper proposed a way for causality-guided continual learning using a soft intervention. The proposed method achieves the SOTA results on different continual learning tasks.

**Strengths:**

* A novel method is proposed.
* New SOTA results on two datasets are obtained

**Weaknesses:**

* Some parts of the paper are not clear. For example, the reviewer does not understand how to obtain equation 4 in the document's main body. Particularly how equation 6 in the appendix is derived from equation 5. This is the paper's key formula, and providing step-by-step derivations or explanations from equations 5 and 6 in the appendix is necessary.
* Some ablation studies are lacking. For example, it would be useful to show the model performance of the contrastive method without O-LoRA (say, full finetuning). Also, model performance of O-LoRA + DA + CGCL is also useful.

**Questions:**

* How is eq6 is derived from eq5?
* How is the performance of the contrastive method without O-LoRA?
* How is the performance of O-LoRA + DA + CGCL?
* Since the sample size is small (500), what is the standard deviation of the methods in Tables 3 and 4? Would be great to provide confidence intervals or conduct statistical significance tests in addition to reporting standard deviations.

---

### Official Review · Reviewer_G885 · 2024-11-04

**Soundness:** 2
**Presentation:** 1
**Contribution:** 2
**Rating:** 5
**Confidence:** 4

**Summary:**

The paper proposes a causality-guided continual learning approach that leverages data augmentation and variational inference to enhance performance in data-limited scenarios. Experimental results demonstrate that this approach outperforms several baseline methods.

**Strengths:**

1. The proposed method is well-motivated and supported by theoretical foundations.
2. It outperforms several baseline methods.

**Weaknesses:**

1. The description of the proposed method lacks detail. For example, regarding the noise added to the weights, did the authors introduce a trainable standard deviation parameter for all weights in LoRA?

2. The computational cost associated with the additional trainable parameters and data augmentation methods should be analyzed.

3. A baseline method using only variational inference should be reproduced and compared to the proposed method.

4. Results combining the proposed method with approaches other than O-LoRA should also be reported.

Minor

5. Lines 698 and 700: The word "set" is misspelled as "se."

**Questions:**

See above.

---

### Official Review · Reviewer_drUq · 2024-11-05

**Soundness:** 3
**Presentation:** 2
**Contribution:** 2
**Rating:** 3
**Confidence:** 3

**Summary:**

This paper studies continual learning for LLMs. The authors propose a method based on causal intervention for continual learning, where the input data $X$ is considered as a confounder between model parameters $\Theta$ and output predictions $Y$, and the learning is guided by a soft causal intervention on $\Theta$. The paper conducts experiments on two base LLMs and a wide range of tasks, demonstrating the effectiveness of the method when the training dataset is small.

**Strengths:**

1. The paper introduces a causal perspective on continual learning, which is new in the LLM continual learning area.
2. The experiments on a wide range of tasks demonstrate the advantages of the proposed method over compared baselines.

**Weaknesses:**

1. The causal graph and causal relations are pre-assumed without clear explanations. Additionally, the definitions of the random variables are a bit confusing. Since you evaluate on classification tasks, is $X$ only the input without the label? If yes, how does the ground truth label $Y$ play a role in the causal graph? Shouldn't it point to $\Theta$ as the model is trained on pairs of $x$ and $y$? Additionally, why do the criteria hold to apply the backdoor theorem? What about other confounders such as linguistic features and world knowledge?
2. I'm not sure if classification tasks are suitable to evaluate continual learning for LLMs. The current LLMs already have these multi-task abilities. For example, what is the zero-shot or few-shot performance of a Llama2-7B-instruct model on these tasks? If an instruction-tuned model can already perform these tasks, it's less meaningful to evaluate on these tasks. Maybe a more realistic setting is to learn newly occurred factual knowledge.
3. The experiment setting is not clear. Particularly, is the O-LoRA + DA baseline using exactly the same augmented as your method? If not, what is the performance of doing data augmentation exactly the same as the proposed CGCL, plus the replay loss term?

**Questions:**

Please see the weaknesses.

---

### Meta-Review · Area_Chair_oTAR · 2024-12-21

**Metareview:**

This paper proposes a causality-guided approach for continual learning in LLMs. The paper introduces a causal intervention framework for continual learning and leverages soft causal intervention to address the issues of catastrophic forgetting and reliance on spurious features. Experiments on two base LLMs and a wide range of tasks demonstrate the superior performance of the proposed method over compared baselines, especially when the amount of data is limited. All reviewers acknowledged the comprehensive experiments and good performance of the method. Reviewers drUq, WY6a, i2KP, and r6CX further appreciated the novelty of the proposed causal framework and method. However, reviewers also raised concerns regarding the validity of the causal graph such as missing important variables (drUq), the importance of continual learning on classification tasks for LLMs (drUq), the comparison with missing baselines and related works (drUq, G885, r6CX, i2KP), the lack of details of the methodology section (G885, WY6a), and the insufficient analysis and ablation study (WY6a, i2KP, r6CX).

In the discussion period, the authors addressed some of these concerns by providing additional comparisons with baselines and ablation studies. The authors also clarified and modified the causal graph and provided additional explanations for the methodology section. Despite these efforts, reviewers maintained the negative ratings and suggested further revision of the writing and more advanced baselines. Therefore, the AC recommends rejecting the paper in its current form.

**Additional Comments On Reviewer Discussion:**

The reviewers’ main concerns include the validity of the causal graph (drUq), the importance of continual learning on classification tasks (drUq), the comparison with missing baselines (drUq, G885, r6CX, i2KP), the lack of details of the methodology section (G885, WY6a), and the insufficient analysis and ablation study (WY6a, i2KP, r6CX). In the discussion, the authors responded with a modified causal graph and justifications for the backdoor theorem. The authors also provided additional experiments comparing with the suggested baselines and conducting the suggested ablation studies. Two of the reviewers (G885, i2KP) acknowledged that their concerns are partially resolved and updated the rating. One reviewer (r6CX) replied but maintained the score. Overall, the reviewers were leaning towards rejection after the discussion.

---

### Decision · Program_Chairs · 2025-01-22

Reject